# Surface Analysis of Chamber Coating Materials Exposed to CF$_4$/O$_2$ Plasma

**Seung Hyun Park, Kyung Eon Kim and Sang Jeen Hong *** 

Department of Electronics Engineering, Myongji University, Yonin 17058, Korea; miline8@naver.com (S.H.P.); kimky1013@naver.com (K.E.K.)
* Correspondence: samhong@mju.ac.kr

**Abstract:** Coating the inner surfaces of high-powered plasma processing equipment has become crucial for reducing maintenance costs, process drift, and contaminants. The conventionally preferred alumina (Al$_2$O$_3$) coating has been replaced with yttria (Y$_2$O$_3$) due to the long-standing endurance achieved by fluorine-based etching; however, the continuous increase in radio frequency (RF) power necessitates the use of alternative coating materials to reduce process shift in a series of high-powered semiconductor manufacturing environments. In this study, we investigated the fluorine-based etching resistance of atmospheric pressure-sprayed alumina, yttria, yttrium aluminum garnet (YAG), and yttrium oxyfluoride (YOF). The prepared ceramic-coated samples were directly exposed to silicon oxide etching, and the surfaces of the plasma-exposed samples were characterized by scanning electron microscopy, energy-dispersive X-ray spectroscopy, and X-ray photoelectron spectroscopy. We found that an ideal coating material must demonstrate high plasma-induced structure distortion by the fluorine atom from the radical. For endurance to fluorine-based plasma exposure, the bonding structure with fluoride was shown to be more effective than oxide-based ceramics. Thus, fluoride-based ceramic materials can be promising candidates for chamber coating materials.

**Keywords:** plasma resistance; inner chamber wall coating; coating materials; atmospheric plasma spraying; yttria

## 1. Introduction

Three-dimensional semiconductors with high aspect ratios have been extensively investigated to achieve high-performance, multifunctional, and miniaturized semiconductor devices, and high-density plasma processes comprising high-powered etching have become essential for the fabrication of dynamic random-access memory capacitors. The etching is conducted by both physical ion bombardment and chemical reaction on a wafer surface; on the other hand, etching in fluorine-based plasma may also cause a chemical reaction between the fluorine plasma and interior coating material of the process chamber, forming fluoride layers and desorption of fluoride particles by physical bombardment [1]. In addition, a recent patent highlights chamber wear [2] and supports the importance of chamber coating in preventive maintenance (PM) as well as process drift caused by worn components inside the plasma chamber. When the chamber wall and components are exposed to a high-density plasma environment over a given time, surface erosion occurs due to the chemisorption and physisorption of fluorinated compounds contaminating the chamber [3]. Therefore, investigation of the etching resistance of ceramic coatings is important for their applications.

In plasma processes, radio frequency (RF) power is used to accelerate ion bombardment to the target. Furthermore, higher RF power leads ion bombardment to be more anisotropic in deep reactive ion etching [4]. When increased RF power is applied for deep reactive ion etching, the wafer surface, as well as the inner plasma chamber surface, become exposed to ion bombardment [5]. As a result, contaminants such as particles form, causing chamber contamination [6]. To alleviate this problem, ceramic coatings

have been developed due to their superior chemical stability and endurance in reactive plasma chemistries. This approach reduces contaminants during the plasma process and, in turn, PM requirements [6,7]. Ceramic coating materials investigated in this research include aluminum oxide (alumina: $Al_2O_3$), yttrium oxide (yttria: $Y_2O_3$), yttrium aluminum garnet (YAG), and yttrium oxyfluoride (YOF). $Al_2O_3$ has been widely adopted in various fields due to its low cost and high mechanical strength, and many ceramic components of plasma equipment are still made from alumina. In the high-density plasma, $Y_2O_3$ has replaced $Al_2O_3$ as a chamber interior coating to increase plasma resistance. Although $Y_2O_3$ shows superior plasma resistance to conventionally employed $Al_2O_3$, the high cost of $Y_2O_3$ requires the use of YAG to accommodate the chemical properties of $Al_2O_3$ and the mechanical properties of $Y_2O_3$. The strongly fluorinated surface of $Y_2O_3$ with the formation of Y–F bonds in electron-evaporated $Y_2O_3$ film was observed after a $CF_4/O_2/Ar$ plasma, and Kim et al. suggested that $YF_3$ can be used as a chamber coating in plasma equipment [7]. For this reason, we also considered YOF as a coating material because it exhibits similar physical and electrical parameters to $YF_3$ [8].

Not only the material type but also the material processing method determines plasma resistance capability. Commercially available coating methods include anodization, aerosol deposition (AD), and atmospheric plasma spraying (APS), and APS was adopted in this work because it is the most common ceramic coating method for components in the semiconductor industry. APS forms relatively thick films in a short period, but it develops undesirable surface roughness and layered structures that cause surface defects, such as pores and microcracks [9]. AD has been studied as a coating technology with excellent uniformity compared to APS, but it still cannot form a coating layer over a thickness of 100 μm [10].

When ceramic materials become exposed to fluorine-based plasma, ion bombardment may affect their surface properties, such as morphology and chemical bonding, and the plasma-induced erosion of the inner surface coating may lead to undesired process drift or increased contamination. Our postulation for this work was as follows: (1) Plasma exposure alters the surface chemical bonding of the ceramic coating; (2) halogen species radicals in the plasma participate in the chemical reaction with ceramic components; (3) this alteration changes the physical properties of the ceramic coating, such as surface hardness, by causing morphological evolution; and (4) the repetitive ion bombardment and radical reactions make the surface structure more vulnerable to deformation. To investigate this hypothesis, we performed a series of plasma exposure experiments on the ceramic-coated samples and analyzed the effects of plasma on material characteristics. To expedite the process, we intentionally located the ceramic samples on a wafer chuck instead of locating them on a chamber inner wall to allow direct plasma irradiation. In addition, we applied 300 W RF bias power to increase ion bombardment on the ceramic samples with an hour of elapsed process time. Chamber damage accumulates over a long period of time through several processes. Therefore, in this experiment, an acceleration experiment causing coating damage by plasma in a short time was performed by applying a large bias power. We finally evaluated the results of atmospheric plasma-sprayed materials with scanning electron microscopy (SEM), energy-dispersive X-ray spectroscopy (EDX), and X-ray photoelectron spectroscopy (XPS).

## 2. Materials and Methods

$Al_2O_3$ has been the most widely used ceramic coating material in semiconductor equipment components due to its mechanical properties, but it reacts with fluoride species in high-powered plasma over a temperature range [11]. $Y_2O_3$ is suggested as an alternative to increase halogen chemical resistance and avoid the formation of aluminum fluoride (AlF) despite the increased material cost. Factors affecting the generation of contamination particles include pores, roughness, microstructure, and characteristics of fluorine compounds. The greater the number of pores in the coating layer is, the weaker the cohesion strength between atoms becomes; thus, the contamination particles are easily detached,

even by weak physical impact [12]. The rougher the surface is, the larger the surface area becomes, making it easier to react. As the etching proceeds along the highly reactive cracks and grain boundaries, the generation of contaminants varies due to differences in the microstructure [12,13]. When fluoride compounds have been formed by the reaction of coating materials and fluorine, the nonvolatile byproduct may become a contaminant source in the wafer production process. The behavior of fluorine plasma chemistry also depends on the type of yttria and the structure of the material composition in plasma etching [14–16].

In this research, we employed four different types of ceramic coating materials, $Al_2O_3$, $Y_2O_3$, and YAG, and YOF, to investigate the effect of fluoride plasma exposure. Atmospheric plasma-sprayed samples were prepared on 5-mm-thick A6061 aluminum alloy substrates ($15 \times 15$ mm$^2$). As the surface roughness of the prepared samples varied, we initially polished their surfaces. Once the surface-polished, pristine samples were prepared, the samples were exposed to fluorinated plasma. In the experiment, 13.56 MHz RF powered inductively coupled plasma-reactive ion etch (ICP-RIE), manufactured by Plasmart, Daejeon, Korea, was used to employ $CF_4/O_2$ plasma. ICP plasma yields higher density plasma than capacitively coupled plasma with more ions and radicals due to its high electron temperature and electron density [17]. The plasma irradiation conditions were an ICP source power of 450 W, $CF_4$ and $O_2$ gas flow of 55 sccm, base pressure of 20 mTorr, and 300 W 12.56 MHz RF bias power. Experimental apparatus including sample preparation is shown in Figure 1.

- *Al* alloy A6061 & Cleaning
- APS coating (*Al₂O₃*, *Y₂O₃*, *YAG*, *YOF*)
- Dicing (15 ×15 mm²) & Polishing
- Plasma Exposure (ICP-RIE)
  - 13.56 MHz 450 watts source power
  - 12.56 MHz 300 watts bias power
  - *CF₄/O₂* at 20 mTorr
  - 1100, 2300, and 3500 seconds
- SEM : Surface morphology
- EXD : Elemental analysis
- XPS : Binding energy

**Figure 1.** Description of sample preparation and experimental conditions.

The samples were directly placed on the wafer chuck to increase the chemical reaction of fluorine with the ceramic surface, and an RF bias of 300 W was applied to accelerate physical ion bombardment onto the sample surface. In ICP-RIE, increased RF bias power positively affects the electron energy distribution, which increases the plasma potential [17]. In conventional semiconductor etching, 300 W bias power with 450 W source power is an exaggerated condition, but this experimental condition is conceptually valid to accelerate ceramic surface erosion. In this experiment, the applied bias power is larger than conventional etching conditions, so erosion can be checked faster without undergoing the entire process. Because the bias power increases, the ion bombardment increases, so the damage to the coating increases as well. To confirm the validity of the applied bias power, numerical simulation was performed for the total wall impact energy and wall flux. Since argon gas is a monoatomic gas, there are few other interactions, unlike oxygen gas. Therefore, it is possible to intuitively check the effect of changes in experimental conditions. In other words, to exclude the influence of other interactions and to confirm that the bias power affects the ion bombardment in the general case, a simulation was conducted using argon gas. Except for the type of process gas, the other experimental conditions were kept the same. In this simulation, the parameters that dominantly influence the ion flux, such as electron number density ($n_e$), ion wall impact energy ($E_i$), and sheath potential ($V_{sh}$), were investigated. Electron number density refers to the number of electrons in a unit volume of

plasma, and it refers to the amount of positive plasma potential ($V_p$) with respect to the ground.

As shown in Figure 2, both the electron number density and sheath potential increase when the bias power increases. As the electron number density of the surface increases, the surface becomes negatively charged, and the surface potential decreases. As the sheath potential ($V_{sh}$) is determined by the difference between the surface potential ($V_f$) and the plasma potential ($V_p$), $V_{sh} = V_p - V_f$, the sheath potential also tends to increase as the bias power increases. In addition, as shown in Figure 3, the ions enter the surface with greater energy as the bias power increases. When the ion strikes the surface, ion wall impact energy ($E_i$) depends on the mass of the ion and the velocity with $E_i = \frac{1}{2}m_i u_i^2$, where $m_i$ is the mass of ion, and $u_i$ is the velocity of the ion incident on the surface. As the ion mass is constant, the velocity is related to the ion wall impact energy. This is influenced by the sheath, and the sheath forms a vertical electric field in the plasma, accelerating the ions incident on the surface [18]. That is, the velocity of the ion incident on the surface increases as the bias power increases. Moreover, the ion velocity in plasma can be expressed as electron temperature ($T_e$) with Bohm velocity, $u_i = \sqrt{eT_e/M_i}$ [19]. The increase in electron temperature indicates an increase in the ion velocity. In addition, the ion velocity also affects the ion flux. As shown in Figure 4, as the electron temperature increases, the ion flux also increases. The increase in the ion velocity affects the increase in the ion flux incident on the surface. In conclusion, as the bias power increases, the ion flux increases. From this, it was concluded that ion bombardment generally increases with increasing bias power. In other words, in a general case, if the bias power is increased, results that meet the existing process conditions can be obtained in a short time. Based on this result, we applied 300 W RF bias power at the bottom to accelerate the plasma erosion of the ceramic-coated samples. The experiments were performed in extended elapsed etching periods of 1100, 2300, and 3500 s. Once the etching was complete, we analyzed the surfaces using SEM, EDX, and XPS.

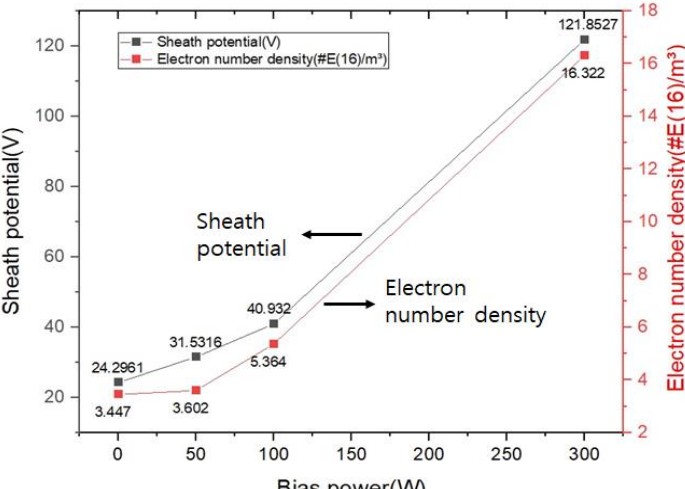

**Figure 2.** Relationship between sheath potential and electron number density with bias power to 1-D plasma simulation using 12.56 MHz bias power.

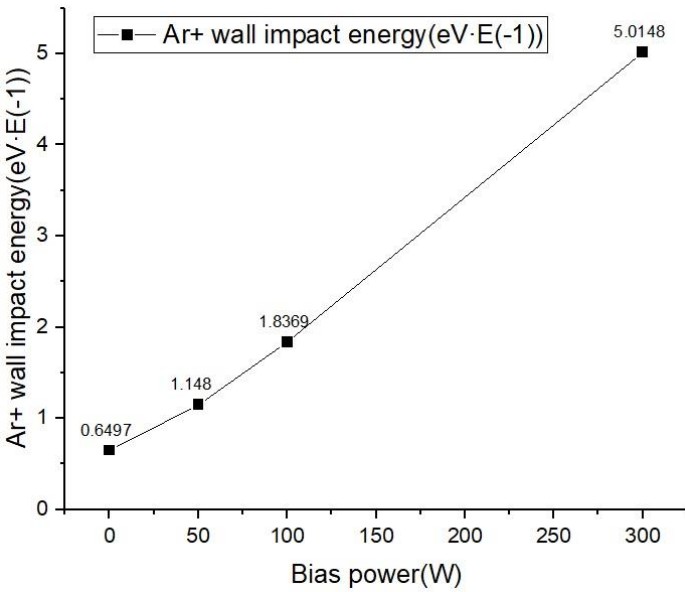

**Figure 3.** Relationship between ion wall impact energy and bias power to 1-D plasma simulation using 12.56 MHz bias power.

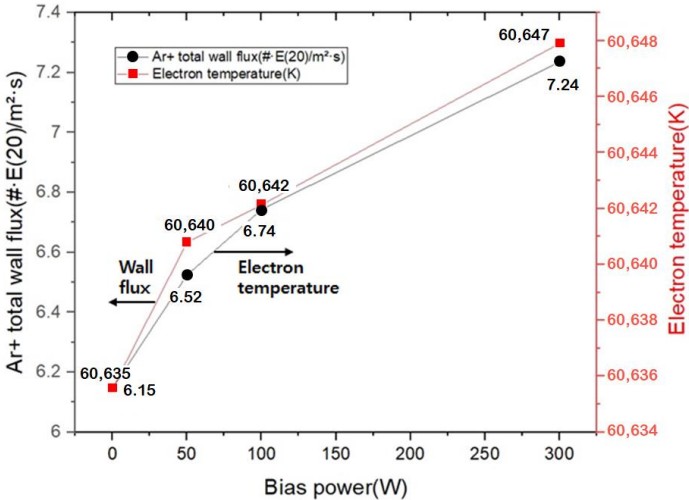

**Figure 4.** Relationship between total wall flux and electron temperature with bias power to 1-D plasma simulation using 12.56 MHz bias power.

## 3. Results and Discussion

### 3.1. Surface Morpholgy

Once the plasma exposure was performed for all the prepared samples, we investigated the surface morphology of the samples by SEM. According to the SEM images in Figure 5, the surface erosion of $Al_2O_3$ appeared at 1100 s, as shown in Figure 5b–d, after plasma exposure, and it gradually increased afterward. $Y_2O_3$ showed good etching resistance throughout the plasma exposure experiment, as presented in Figure 6. It is possible that the pores which appear in Figure 6c are somewhat larger than those of the pristine samples, but this is not obvious. For the YAG sample shown in Figure 7, we observed a unique surface morphology. Grain boundaries were observed from 1100 s of etching, as shown in Figure 7a, and the material loss along the grain boundaries was much more apparent as the plasma exposure time increased, as shown in Figure 7b,c. YOF in Figure 8 showed the lowest amount of plasma-induced damage on the surface. The surface did not seem damaged until the process proceeded for 2300 s. Although the pristine YOF was not polished to a mirror finish, the SEM image shown in Figure 8d is not significantly different

from that of the pristine YOF. In morphological analyses, samples that included $Al_2O_3$ showed noticeable damage from plasma exposure. Conversely, samples including $Y_2O_3$ had superior plasma-induced damage resistance compared with other samples. Surface morphology is one way to visually inspect plasma erosion on the ceramic surface, but it is not straightforward to determine the plasma resistance of the prepared samples. It might relate to their chemical composition and chemical bonding, which may have endowed them with resistance to plasma exposure; thus, we also investigated their surface chemistry with EDX.

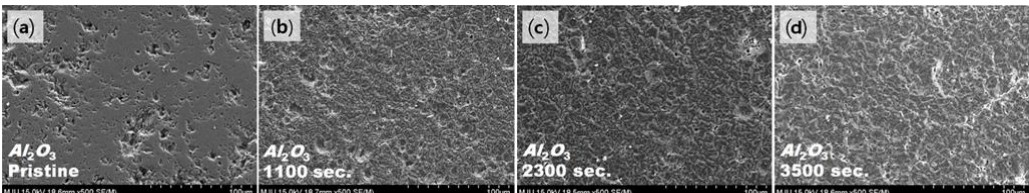

**Figure 5.** SEM images of $Al_2O_3$ samples: (**a**) pristine $Al_2O_3$, (**b**) $Al_2O_3$ etched for 1100 s, (**c**) 2300 s, and (**d**) 3500 s.

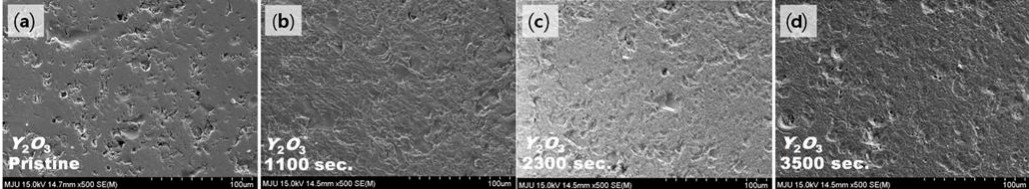

**Figure 6.** SEM images of $Y_2O_3$ samples: (**a**) pristine $Y_2O_3$, (**b**) $Y_2O_3$ etched for 1100 s, (**c**) 2300 s, and (**d**) 3500 s.

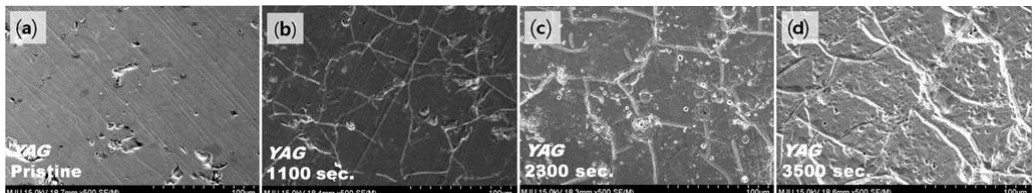

**Figure 7.** SEM images of YAG samples: (**a**) pristine YAG, (**b**) YAG etched for 1100 s, (**c**) 2300 s, and (**d**) 3500 s.

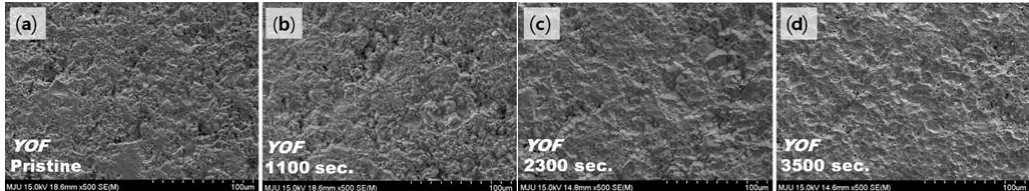

**Figure 8.** SEM images of YOF samples: (**a**) pristine YOF, (**b**) YOF etched for 1100 s, (**c**) 2300 s, and (**d**) 3500 s.

### 3.2. Surface Composition

Morphological analyses showed that plasma-induced damage resistance for the four types of samples showed different behaviors, and SEM observation is related to material composition and chemical bonding. Thus, we investigated their surface chemical compositions with EDX, as summarized in Figure 9. A significant change in the weight proportions of Al and O was detected throughout the plasma exposure, but the fluorine peak was detected at 3500 s of etching of $Al_2O_3$. $Al_2O_3$ becomes vulnerable to plasma ion bombardment within a short period, and fluorine atoms can easily be captured in the crater of the surface of $Al_2O_3$. When $Al_2O_3$ samples were exposed to the fluorine-based plasma, radials that include the fluorine atom collide with the $Al_2O_3$ surface. After this, fluorine

atoms from radicals react with $Al_2O_3$. In this sequence, AlF is formed on the $Al_2O_3$ surface. However, it may not have enough binding energy with original materials to endure continuous ion bombardment. If the fluorine residue is formed on the surface, the continuous ion bombardment might remove the unbounded fluorine atoms until a deeper crater is formed. We cannot overlook the possibility that a small number of fluorine atoms residing on the surface might evaporate before and during the measurement; however, $Al_2O_3$ is vulnerable to plasma-induced damage, especially in the case of a large RF bias power.

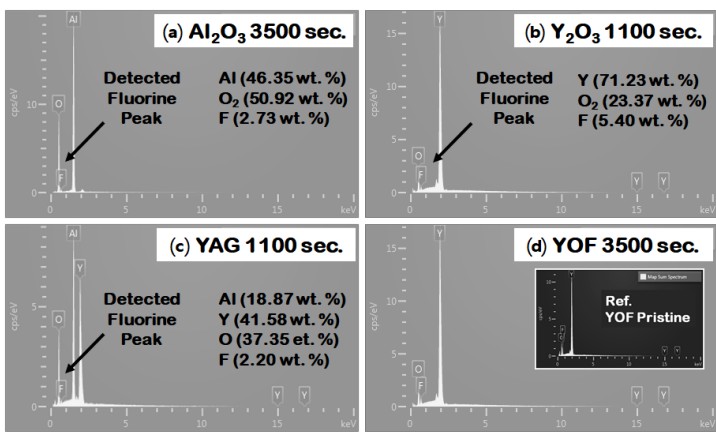

**Figure 9.** EDX spectra of (**a**) $Al_2O_3$ etched for 3500 s, (**b**) $Y_2O_3$ for 1100 s, (**c**) YAG for 1100 s, and (**d**) YOF for 3500 s.

In the case of $Y_2O_3$ samples, the evidence of fluorine appeared for an etching time of 1100 s, and 3.5–6.3 wt.% fluorine was detected at 3500 s. Although the appearance of fluorine might be initiated from the beginning of the plasma exposure, we had fewer concerns about the formation of $YF_3$ because of the high material density of $Y_2O_3$ and the large amount of energy required for forming Y–F bonds. The surface morphology shown in Figure 6d also supports our understanding. Therefore, we consider that $Y_2O_3$ is superior to $Al_2O_3$ in terms of fluorine-based plasma etch resistance. The appearance of the fluorine atom of YAG samples started with 2300 s of etching with a small weight percent (around 2.2%), as shown in Figure 9c. No significant fluorine atom detection was noted for YAG, but we already observed a unique material loss of YAG samples along the grain boundaries, as presented in Figure 7. We also evaluated the delaminated coating sample in the plasma exposure experiment. In this sequence, the weight ratio of the Al atom steadily decreased throughout the samples. This suggested that the damage in the grain boundaries was related to the vulnerability of $Al_2O_3$. On the contrary, the grain itself showed superior plasma-induced damage resistance to the grain boundary, which is related to $Y_2O_3$. The high-Y region had less plasma-induced erosion than other regions, indicating that $Y_2O_3$ had higher durability to plasma exposure than $Al_2O_3$. The consideration of YOF arose from the higher plasma resistance, but the absorption of fluorine may vary with surface pretreatment. Thus, we intentionally introduced fluorine into the material composition before exposure to plasma.

Previously, we anticipated the potential of YOF by the changes on the sample surface, as shown in Figure 8, and the EDX results gave a similar chemical composition on the surface, as shown in Figure 9d. We observed a weight percent loss of oxygen by 3–5 wt.% until 2300 s of etching and a weight percent gain of fluorine by 5%, but it was soon saturated after 2300 s. This implied that the YOF samples were barely damaged by plasma exposure. Even though $Y_2O_3$ had superior resistance to plasma-induced damage, YOF samples showed even higher durability against plasma than the others because of the bonding with fluorine. The YOF sample had already reacted with the fluorine atom, so it could preserve its original structure upon plasma exposure.

EDX data support the notion that the material including $Y_2O_3$ had superior durability to plasma exposure. In YAG samples, most $Al_2O_3$ erosion occurred in a specific area

that included fewer Y atoms than the others. Although YOF samples showed that the weight ratio of fluorine and oxygen atoms was changed, morphological and chemical composition changes were not noticeable. In other words, YOF showed the highest plasma-induced damage resistance. However, morphological and chemical composition analyses alone could not explain these phenomena. Therefore, XPS was performed to reveal the relationship between these results and chemical bonding.

### 3.3. Chemical Bonding

$Al_2O_3$ and YAG had noticeable Al atom loss when they were exposed to fluorine-based plasma for 3500 s. When they were exposed to the plasma, the fluorine atom from the radical including fluorine affected Al–O bonding to combine with fluorine. The binding energy of Al–O bonding increased rapidly due to the combination of the fluorine atom in Figures 10b and 11b. Furthermore, $Al_2O_3$ had an additional binding energy shift, as shown in Figure 10c, meaning that $Al_2O_3$ had a rapid chemical reaction with fluorine-based radicals, and this sequence proceeded explosively for over 3500 s. Although YAG showed less binding energy shift in Figure 11b,c, YAG also had noticeable chemical bonding alternation for over 3500 s. These phenomena could explain that Al–O bonding in $Al_2O_3$ and YAG combined rapidly with the fluorine atom from plasma and then generated AlF that would be etched in a short time, as shown by EDX. In short, the rapid change in Al–O bonding in Al2p peaks supported the assumption that fluorination in $Al_2O_3$ and YAG proceeded swiftly and the fluorinated layer was etched simultaneously.

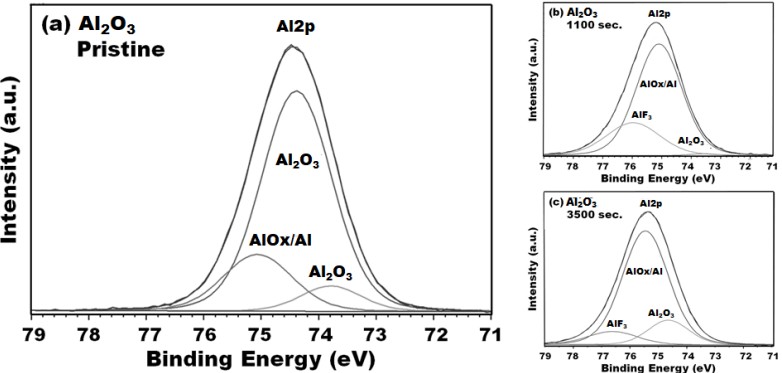

**Figure 10.** XPS spectra of $Al_2O_3$: Al2p peaks in (**a**) pristine $Al_2O_3$, (**b**) $Al_2O_3$ etched for 1100 s, and (**c**) $Al_2O_3$ etched for 3500 s.

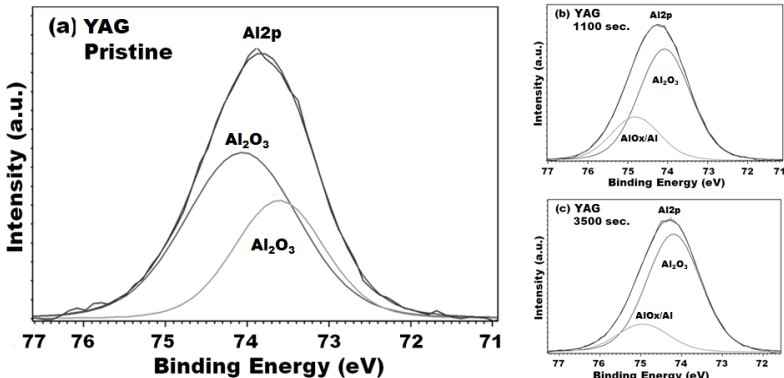

**Figure 11.** XPS spectra of YAG: Al2p peaks in (**a**) pristine YAG, (**b**) YAG etched for 1100 s, and (**c**) YAG etched for 3500 s.

$Y_2O_3$ and YOF showed superior durability to plasma exposure. In XPS spectra, peaks in Y3d showed a similar trend that had less peak shifting, indicating that fluorination generated fluorinated layers on $Y_2O_3$ and YOF but proceeded more slowly than those

on Al$_2$O$_3$ and YAG. However, Y3d peaks alone could not prove the superior durability of Y$_2$O$_3$ and YOF to fluorine-based plasma exposure. To authenticate their performance, especially the performance of Y$_2$O$_3$, we investigated the O 1*s* spectra. In Figure 12a,c, although pristine Y$_2$O$_3$ was fluorinated by fluorine-based plasma, O 1*s* peaks in Y$_2$O$_3$ had small binding energy shifting compared with Figure 12a,c. In addition, Figure 12b,c did not show any binding energy shifting or bonding changes. In other words, the information in O 1*s* implied that Y$_2$O$_3$ did not undergo a dramatic chemical bonding change for over 3500 s. This result could support the theory that Y$_2$O$_3$ had superior resistance to plasma exposure, evidenced by the minor morphological changes and chemical compositional changes in long-term plasma exposure.

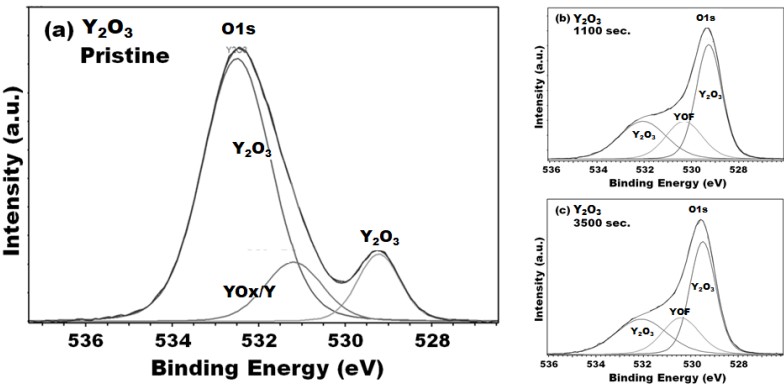

**Figure 12.** XPS spectra of Y$_2$O$_3$: O1s peaks in (**a**) pristine Y$_2$O$_3$, (**b**) Y$_2$O$_3$ etched for 1100 s, and (**c**) Y$_2$O$_3$ etched for 3500 s.

Y3d in YOF also suggested that YOF had little bonding distortion upon plasma exposure. Y3d peaks showed that YOF sustained its yttrium-based bonding, even after long-term plasma exposure. As a result, YOF reacted less, and even fluorination proceeded, as shown in Figure 9d. Therefore, further XPS analysis is required to verify its stability in fluorine-based plasma. The O1s peak in Figure 13a,c also showed less binding energy shifting, as for Y$_2$O$_3$. Less binding energy shifting meant that the original bonding had little alteration as much more energy was required to distort its chemical structure. In addition, Figure 13b,c illustrated that the chemical bonding in O1s did not change, even after 2400 s. This result showed that YOF could resist chemical modification by long-term plasma exposure. As a result, YOF could be an excellent chamber coating material. XPS analyses support the results from morphological observations and EDX. Rapid morphological changes and significant chemical composition changes may be related to chemical bonding distortion. When chemical bonding is altered by fluorination, more shifted bonding would lead to overall chemical bonding, which would collapse more easily than less changed ones. In summary, superior coating materials guarantee less chemical bonding changes in plasma exposure.

Compared with a similar experiment with aerosol-deposited Y$_2$O$_3$ and YF$_3$-coated samples in NF$_3$ plasma [1], we observed a smaller amount (5.40 wt.%) of fluorinated Y$_2$O$_3$ with the APS coating method. Very recent research on the erosion behavior of Y$_2$O$_3$ in fluorine-based etching plasma reported that oxide ceramic materials such as Y$_2$O$_3$ were eroded by a physiochemical mechanism, and it revealed that the material orientation is also related to the surface plasma interaction [20]. The limitation of this research is the absence of the material hardness [21] and electrical breakdown analysis [22], which can be good representations of the endurance of the material from the plasma ion bombardment and the prevention of the plasma arcs to create potential contamination, respectively. However, the crucial factor in this research is the aspect of plasma condition with the increased RF bias power, and the material hardness and electrical breakdown remain to be considered in future works.

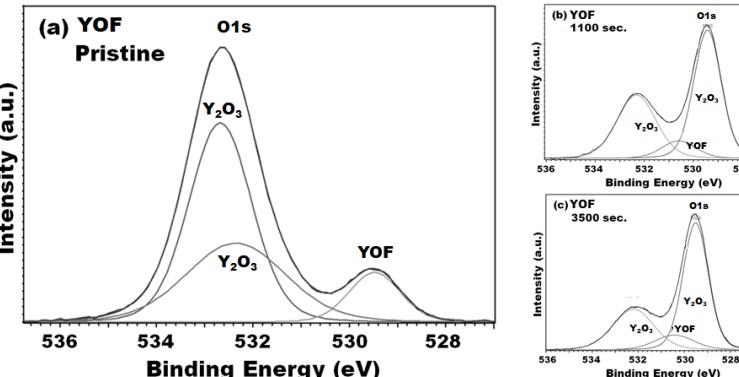

**Figure 13.** XPS spectra of YOF: O1s peaks in (**a**) pristine YOF, (**b**) YOF etched for 1100 s, and (**c**) YOF etched for 3500 s.

## 4. Conclusions

Plasma-induced erosion on ceramic coating materials related to fluoride generation tendency and fluoride etching tendency have been investigated. When fluoride is generated on the surface, it breaks metal oxide bonding, such as Al–O and Y–O. This phenomenon made the ceramic surface vulnerable to ion bombardment. When the ceramic coating material was exposed to the fluorine-based plasma, its original bonding became weak or was damaged by fluoride. Weakened bonding and fluoride became easily etched by ion bombardment. In addition, RF bias power did not noticeably increase the ion density in the plasma above a bias power of 50 W in our simulation study. However, increased ion density led to faster fluoride etching.

An ideal coating material must show low plasma-induced damage. Thus, chemical bonding stability should be maintained to minimize plasma-induced damage. When chemical bonding became unstable, it became eroded more easily by ion bombardment. Although fluoride affected plasma-induced damage, it might protect the original bonding structure, such as in YOF. Therefore, fluoride-based metal oxide, especially yttrium-based oxides such as yttrium fluoride, could be effective ceramic-based chamber coating materials against plasma-induced damage.

**Author Contributions:** Conceptualization, S.J.H.; experiment and analysis. S.H.P.; writing—original draft and revision, S.H.P., S.J.H., and K.E.K.; software, K.E.K.; review and editing, S.J.H.; funding acquisition, S.J.H. All authors have read and agreed to the published version of the manuscript.

**Funding:** This research was funded by Gyeonggido R&D grant (G14AICT06T10001) and the Korea Institute of Advanced Technology (KIAT) grant funded by the Korea Government (MOTIE) (P0008458, the Competency Development Program for Specialist).

**Institutional Review Board Statement:** Not applicable for studies not involving humans or animals.

**Informed Consent Statement:** Not applicable for studies not involving humans.

**Data Availability Statement:** The data presented in this study are available on request from the corresponding author. The data are not publicly available due to the restriction of the material supplier.

**Acknowledgments:** The authors are grateful to Jung at KoMiCo for providing materials and his valuable comments on the coating technology.

**Conflicts of Interest:** The authors declare no conflict of interest.

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
