# Peer review of "Surface Analysis of Chamber Coating Materials Exposed to CF4/O2 Plasma"

_coatings, doi:10.3390/coatings11010105_

Round 1

Reviewer 1 Report

The paper “Surface Analysis of Chamber Coating Materials Exposed to CF4/O2 Plasma”, by Seung Hyun Park, Kyung Eon Kim and Sang Jeen Hong, present interesting results for coatings domain by using and RF CF4/O2 plasma spay method.

The paper is recommended for publication with minor revisions:

  1. In section 2, the paragraph between Line 88 -Line 97 should be inserted in the Introduction section.
  2. Line 101- please use superscript for “mm2”.
  3. Please explain the following sentence from lines 99-100 “Four different atmospheric plasma-sprayed samples were prepared on 5-mm thick A6061 aluminum alloy substrates (15 × 15 mm2).” What does it mean “atmospheric”? because in the following lines you mentioned the pressure at 20mTorr, that means around 27 mbar, which is bellow atmospheric pressure. You refer that the experiments were performed in the Ambiental atmosphere? Even so, the total pressure of 20mTorr is too low to be considered at atmospheric pressure.
  4. Lines 103 -105 please re-formulate the sentence “In the experiment, 6-inches….”. The sentence is too long and is not clear.
  5. In section 2, the authors mentioned a validation method based on simulation. In the simulation, the authors have used Argon, but in the experiment, they have used CF4/O2. Which is the accuracy of the simulation, because Argon it’s a monoatomic gas with a high sputtering rate, where, in the plasma form, the interactions between species are not so many. In the case of using oxygen plasma (oxygen it's a molecular gas), the situation is different. How do you explain the correlation between Ar simulation and CF4/O2 used for the experiment?
  6. Line 140- please correct the word “plasms”.
  7. Line 154- please correct the word “plasms”.

Author Response

Authors are grateful to reviewers for the valuable comments to improve the quality of the manuscript. We tried all the necessary questions and comments raised from the first round review, and more detailed answers and revisions are in the attachment. Please find the attachment for more details.

Reviewer 2 Report

It is manuscript showing the effect of CF4/O2 plasma on the ceramic-coated samples.

1) Line 37. RF must be defined at the first mention

2) There is no information about modes and equipments for APS of the coatings. Additional information about thickness of the coatings is needed.

3) There is no information about surface roughness after polishing. 

4) What can you tell about the cohesive strength of the coating after treatment? 

Does the microhardness of the coatings change? Please, discuss it. 

Author Response

(The authors gave the same response as above.)

Reviewer 3 Report

The work presented by the authors are good. Some major modifications are required for final publication:

  1. Novelty statement is missing from the abstract.
  2. 2. Literature review part is very week.
  3. 3. Its better to include one flow chart in materials and methods section.
  4. Please compare the results with existing literature.
  5. conclusion part is also not satisfactory.

Author Response

(The authors gave the same response as above.)

Round 2

Reviewer 2 Report

The work presented by the authors are normal.

Reviewer 3 Report

Article is ready for publication.